# A New Formulation for Strigolactone Suicidal Germination Agents, towards Successful *Striga* Management

**DOI:** 10.3390/plants11060808

**Published:** 2022-03-18

**Authors:** Muhammad Jamil, Jian You Wang, Djibril Yonli, Rohit H. Patil, Mohammed Riyazaddin, Prakash Gangashetty, Lamis Berqdar, Guan-Ting Erica Chen, Hamidou Traore, Ouedraogo Margueritte, Binne Zwanenburg, Satish Ekanath Bhoge, Salim Al-Babili

**Affiliations:** 1The BioActives Lab, Center for Desert Agriculture, King Abdullah University of Science and Technology, Thuwal 23955-6900, Saudi Arabia; muhammad.jamil@kaust.edu.sa (M.J.); jianyou.wang@kaust.edu.sa (J.Y.W.); lamis.berqdar@kaust.edu.sa (L.B.); guanting.chen@kaust.edu.sa (G.-T.E.C.); 2Institut de l’Environnement et de Recherches Agricoles (INERA), Ouagadougou 04 BP 8645, Burkina Faso; d.yonli313@gmail.com (D.Y.); hamitraore8@yahoo.com (H.T.); margoued616@gmail.com (O.M.); 3UPL House, Express Highway, Bandra-East, Mumbai 400 051, Maharashtra, India; rohit.patil2@upl-ltd.com (R.H.P.); bhogese@upl-ltd.com (S.E.B.); 4International Crops Research Institute for the Semi-Arid Tropics (ICRISAT), Niamey BP 12404, Niger; m.riyazaddin@cgiar.org (M.R.); p.gangashetty@cgiar.org (P.G.); 5Plant Science Program, Biological and Environmental Science and Engineering Division, King Abdullah University of Science and Technology (KAUST), Thuwal 23955-6900, Saudi Arabia; 6Institute for Molecules and Materials, Radboud University, 6525 AJ Nijmegen, The Netherlands; b.zwanenburg@science.ru.nl

**Keywords:** *Striga hermonthica*, seedbank, suicidal germination, strigolactone analogs, witch weeds, methyl phenlactonoate

## Abstract

*Striga hermonthica*, a member of the *Orobanchaceae* family, is an obligate root parasite of staple cereal crops, which poses a tremendous threat to food security, contributing to malnutrition and poverty in many African countries. Depleting *Striga* seed reservoirs from infested soils is one of the crucial approaches to minimize subterranean damage to crops. The dependency of *Striga* germination on the host-released strigolactones (SLs) has prompted the development of the “Suicidal Germination” strategy to reduce the accumulated seed bank of *Striga*. The success of aforementioned strategy depends not only on the activity of the applied SL analogs, but also requires suitable application protocol with simple, efficient, and handy formulation for rain-fed African agriculture. Here, we developed a new formulation “Emulsifiable Concentration (EC)” for the two previously field-assessed SL analogs Methyl phenlactonoate 3 (MP3) and Nijmegen-1. The new EC formulation was evaluated for biological activities under lab, greenhouse, mini-field, and field conditions in comparison to the previously used Atlas G-1086 formulation. The EC formulation of SL analogs showed better activities on *Striga* germination with lower EC_50_ and high stability under Lab conditions. Moreover, EC formulated SL analogs at 1.0 µM concentrations reduced 89–99% *Striga* emergence in greenhouse. The two EC formulated SL analogs showed also a considerable reduction in *Striga* emergence in mini-field and field experiments. In conclusion, we have successfully developed a desired formulation for applying SL analogs as suicidal agents for large-scale field application. The encouraging results presented in this study pave the way for integrating the suicidal germination approach in sustainable *Striga* management strategies for African agriculture.

## 1. Introduction

*Striga hermonthica*, an obligate root-parasitic weed, is one of the major biotic constraints to cereal production in sub-Saharan Africa [1,2,3,4]. After spreading in about 32 African countries, *Striga* has infested about 50 million hectares of arable land in Africa [5,6]. The crop yield losses due to *Striga* infestation can vary from 40 to 100%, causing annual losses of around US $10 billion and threatening the life and food security of 300 million African people [7,8,9,10,11]. Developing suitable *Striga* control strategies is crucial for not only reducing the extent of damage but also retaining further spread into the non-contaminant fields [12]. However, the management of *Striga* is challenging due to the production of ~0.1 million seeds per plant [13], up to 20 years of seed longevity [14], complex life cycle [3], underground damage [15], and host dependency of *Striga* seed germination [16]. 

The germination of *Striga* seed requires favorable hot and humid conditions [17] and, most importantly, the perception of host-released chemical signals, such as strigolactones (SLs) [18,19,20]. The germinated *Striga* seeds should establish a connection to the root system of the host to survive due to very limited food reserve in tiny seeds for a short period of time [3,21]. This essential step of *Striga* life cycle leads to a basis for a promising control strategy, known as “Suicidal Germination” [22,23]. The germination of *Striga* seeds can be triggered in the bare soil by direct application of synthetic germination stimulants [24,25]. This germination in the host’s absence is lethal for *Striga*, which can be exploited as a tool to eliminate the accumulated *Striga* seed bank [22,23,26]. Although this concept has been proposed in a number of previous studies [22,23,26,27], but the availability of simple, easy-to-synthesize, and affordable synthetic SL analogs with suitable formulation and application under natural, hot, and rain-fed field conditions is still arguable [28]. Indeed, the selection of suitable germination stimulants, application protocol, and appropriate formulation for field application remain as challenging barriers to the success of this technology. 

Several SL analogs and mimics were developed over the last three decades, but the problem of their efficacy, stability, and synthesis remains unsolved [29,30,31]. To identify suitable suicidal germination stimulants, a few SL analogs have been screened during the past 10 years based on the bioactivity of *Striga* germination under different conditions [25,32,33,34]. Indeed, two potent SL analogs, namely, MP3 and Nijmegen-1, were selected to be tested for suicidal germination activity under field conditions [23,32]. Another crucial step was to devise an appropriate protocol for successful field application. Keeping in mind that scarcity of water, type of soil, frequency and concentration of application, a rain-fed based suicidal application protocol has been proposed recently [23]. In addition, the development of active and viable formulation, suitable for harsh African agricultural conditions, is very crucial for the success of suicidal technology. Although the use of SL analogs formulated in Atlas G-1086 (AG), by Coroda Crop Care, The Netherlands, has been reported in previous studies, this emulsifier is expensive and time-consuming for preparation. Moreover, the solubility problems of active ingredients in the AG formulation and side effects on the host crop have been observed. Alternately, another formulation “Emulsifiable Concentrate (EC)” was developed by the UPL, India. The present study is carried out to test and evaluate the efficacy of this newly developed EC formulation of two very simple and efficient SL analogs under laboratory, greenhouse, and field conditions. We also compared the old AG formulations of both SL analogs on the suicidal germination activity with this new formulation. Besides, this is the first SL analogs formulation as suicidal agents, synthesized on a large-scale for real field application. This EC formulation is user-friendly, handy, water-soluble, highly compatible with SL analogs, and stable at room temperature (>2 years). To this end, our results will lead to developing a package of suicidal technology to combat *Striga* in Africa.

## 2. Results

### 2.1. Striga Seed Germination in Response to EC and AG Formulations of SL Analogs under Lab Conditions

Structure of SL analogs and packing of Emulsifiable Concentrate (EC) formulation of MP3 27EC and Nijmegen 34EC are shown in Figure 1. The scheme of the experiment conducted for *Striga* seed germination bioassays is depicted in Figure 2A. The EC formulation of SL analogs MP3 27EC and Nijmegen 34EC induced about 56–59% *Striga* seed germination at a concentration of 1.0 µM, which was about 3–13% higher than AG formulation (49–57%) (Figure 2B). We also compared the activity of the two EC formulated analogs MP3 27EC and Nijmegen 34EC and observed both analogs at 1, 0.1, and 0.01 µM had similar activity, while Nijmegen 34EC at lower 0.001 µM showed a better activity and exhibited the lower EC_50_ value (0.008 vs. 0.036 µM) (Figure 2C). Importantly, EC formulation effectively reduced the value of EC_50_ in comparison to AG formulation (Figure 2C). Moreover, both EC formulated SL analogs induced seed germination of various *Striga* batches collected from Kenya, Niger, and Burkina Faso. Seed collected from Kenya showed about 9–15% *Striga* seed germination, followed by Burkina Faso batch with 11–13%, and Niger batch with 8–11% germination (Figure 3). The *Striga* seeds collected from Sudan appeared very active, showing maximum germination (~60%) as compared to seed populations collected from Burkina Faso and Niger (8–11%). However, these outcomes revealed that EC formulation of both SL analogs are still able to induce germination of various ecotypes of *Striga* seeds throughout African countries, depending upon seed viability and dormancy.

### 2.2. Stability of EC and AG Formulations of Strigolactone Analogs

Fresh preparations of 1.0 µM SL analogs MP3 and Nijmegen with both formulations demonstrated about 52–64% *Striga* germination. EC formulation exhibited 60–64% germination when compared to AG formulation 52–57% (Figure 4). MP3 in EC formulation remained active even at 12 weeks after application, showing up to 51% *Striga* seed germination; while the AG formulation of MP3 only had a 29% activity on *Striga* germination on week 12. Surprisingly, Nijmegen in AG formulation completely lost its activity in week 6 (~30% *Striga* germination on Week 4); whereas EC formulation of Nijmegen was much more stable and remained active up to week 6 with ~29% *Striga* germination (Figure 4). 

### 2.3. Effect of Various Formulations of Strigolactone Analogs on Striga Emergence in Pots under Greenhouse Conditions

Both formulations of the two SL analogs (MP3 and Nijmegen) were further evaluated by applying at 1.0 µM concentration to *Striga* infested pot under greenhouse conditions (Figure 5A,B). Intriguingly, EC formulation of both analogs showed around 89–99% decrease of *Striga* emergence, which was about 3–5% higher than the reduction detected with the AG formulation (85–96%) (Figure 5C). However, no clear difference was obtained between the two formulated Nijmegen groups. Among the two EC formulated analogs, MP3 27EC had a larger decline in *Striga* emergence (99%) than Nijmegen 34EC (89%). In addition, a higher reduction in *Striga* emergence led to a better growth of the host plant, indicated by an increase of 61–99% in plant height of the host crop (Figure 5D).

### 2.4. Effect of Various Formulations of Strigolactone Analogs on Striga Infection under Mini-Field Conditions

To fulfill the practical purpose of developing EC formulation for alleviating *Striga*, the efficacy of the two SL analogs MP3 27EC and Nijmegen 34EC was tested under mini-field conditions (Figure 6A–C) in INERA, Burkina Faso and ICRISAT, Niger. At INERA, although a huge variation among replications resulted in a non-significant impact on *Striga* germination (Figure 6D), apparently, we observed more *Striga* germination upon the application of MP3 27EC and Nijmegen 34EC (at 1.0 µM) compared to AG formulation. EC formulation of both SLs showed about 15–31% *Striga* germination and the Nijmegen 34EC appeared more active than MP3 27EC (Figure 6D). In addition to *Striga* germination, EC formulation of both SL analogs showed 9–23% reduction in *Striga* emergence, whereas we did not observe any reduction in *Striga* emergence after AG formulation treatment (Figure 6E). Similarly, we noticed a huge variation among replicated mini-boxes from ICRISAT, Niger, making the results non-significant. However, we still observed 35–53% reduction in *Striga* emergence by EC formulation of MP3 27EC and Nijmegen 34EC which was better than the AG formulation (16–21%). Additionally, Nijmegen 34EC showed better activity than MP3 27EC (Figure 7).

### 2.5. Effect of EC and AG Formulation of Strigolactone Analogs on Striga Infection under African Field Conditions

Finally, we investigated the efficacy of MP3 27EC and Nijmegen 34EC under naturally infested farmer-field conditions in INERA, Burkina Faso (Figure 8). In the pearl millet field, we observed 47–60% reduction in *Striga* emergence upon AG formulation of the two SL analogs application compared to blank treatment (Figure 8C). The EC formulation of Nijmegen 34EC showed about 25% reduction in *Striga* emergence in comparison to blank. Surprisingly, EC formulation of both SL analogs had a positive impact on pearl millet grain yield (126–137%). Moreover, MP3 in EC formulation showed a significant increase in grains per panicle over blank treatment (Appendix A). In the sorghum farmer field, we observed 70–87% reduction in *Striga* emergence by SL analogs in AG formulation while EC formulation of MP3 and Nijmegen showed 35–65% reduction in *Striga* emergence (Figure 8E). This reduction in *Striga* emergence revealed a close association with *Striga* biomass, collected at final harvest from sorghum field (Appendix A). We also observed 56–76% reduction in *Striga* biomass over blank treatment in AG formulated SL analogs while the reduction in EC formulated plots was 62–68% over blank treatment (Appendix A). EC formulation treated plots of both SL analogs showed an increase in sorghum grain yield (164–216%) (Figure 8F). AG formulated MP3 showed considerable increase in yield components of sorghum compared to blank treatment (Appendix A). Likewise, in ICRISAT, Niger, we observed a 91% reduction in *Striga* emergence by EC formulated MP3 27EC which was 27% higher than AG formulation of MP3 (66%) (Figure 9). Although MP3 27EC showed a better activity than Nijmegen 34EC, we observed a huge variation in *Striga* infestation among the plots of each treatment. 

## 3. Discussion

*Striga*, an obligate parasitic plant, attaches to the root system of most cereal crops in Africa while its subterranean nature of damage has made its control very difficult [5,35]. Developing suitable control strategies to minimize these underground losses has been proposed and advocated during the past few decades [28,36]. However, the underground damage of the host crop can be decreased only by reducing the seed bank density of *Striga* in the infested soil [37]. The “Suicidal Germination” technology has been suggested and tested in some field studies but with a lot of obstacles and limited success [38,39]. We have been working for the last five years to overcome the challenges of suicidal technology. We found two potent SL analogs MP3 and Nijmegen that can be applied into the field as suicidal agents [8,9]. In addition to the simple and efficient germination stimulants, the selection of suitable formulation is critical to facilitate large-scale field application [23]. A good formulation of SL analogs not only enhances the efficacy but also increases the stability and shelf-life after application into the field [22,40]. Atlas G-1086, a polyoxyethylene sorbitol hexaoleate mixed in cyclohexanone, has been used to formulate SL analogs in some past studies [23,25,26]. Previously, this emulsifier had been used to formulate Nijmegen-1 to apply in tobacco fields parasitized by *Orobanche ramosa* [22,41]. Moreover, some other potent SL analogs were also formulated with AG and used in *Striga* infested pearl millet and sorghum farmers field in Burkina Faso, showing a promising impact on reduction in *Striga* emergence [23,25]. However, there were few drawbacks of using AG to formulate SL analogs. We have been working closely with our partner, UPL, India, on developing suitable, handy, effective, and easily accessible formulated SL analogs that confer not only stability but also the ability to deplete easily of the seed bank in the infested soil. For this purpose, a new formulation “Emulsifiable Concentrate (EC)” of MP3 and Nijmegen (Figure 1) has been prepared by UPL, India.

In this report, we investigated the newly developed EC formulation of MP3 and Nijmegen with in vitro lab bioassays studies and the results indicated that MP3 and Nijmegen would be the potent SL analogs on seed germination (Figure 2). Both analogs with EC formulation showed better induction of *Striga* germination and EC_50_ over AG formulation. The induction of germination of various batches of *Striga* seeds by EC formulation of both SL analogs suggested that this formulation can be equally effective against all ecotypes of *Striga* in various parts of Africa (Figure 3). However, efficacy of SL analogs for both formulations varied for different *Striga* seed populations, with highest germination (60%) in Sudan batch and lowest germination (11%) in Niger batch. This variation among various populations might be attributed to seed viability, dormancy, as well as response to germination stimulants. The bioactivity of EC formulated MP3 for a longer period of time also indicates its potential as a good suicidal agent for field application in African soil. The EC formulated SL analogs possess longer shelf life and stability that is the desired characteristics for real field applications (Figure 4). In our previous studies, we observed several adverse effects of AG formulation on plant development and growth, which can be overcome by the EC formulation. Moreover, it is hard to dissolve SL analogs in AG formulation as compared to EC formulation. We also validated our lab outcome in a pot study under greenhouse conditions. Both SL analogs with EC and AG formulation have shown 85–99% decrease in *Striga* emergence (Figure 5).

However, we failed to get a significant impact on the reduction of *Striga* emergence under mini-field conditions both at INERA, Burkina Faso and ICRISAT, Niger (Figure 6). In fact, we had only one application in the mini-field, which might not have been enough to successfully reach to the *Striga* seeds to induce germination. In addition, a high variation among mini-fields might be due to several reasons, including high *Striga* density in the infested soils, low efficacy of applied treatments, soil type, leakage of water along the side of mini-fields, insufficient gap between application time, and host planting. It is recommended to enhance the number of applications in the mini box (>1) and to give enough time for suicidal death after *Striga* germination. Indeed, when we increased the number to six applications under naturally infested farmer fields in Burkina Faso, we observed very encouraging results, particularly in the sorghum field (Figure 8). Surprisingly, AG formulated SL analogs showed 70–87% reduction in *Striga* emergence in sorghum field and 47–60% in pearl millet field while EC formulated SL analogs did not reveal any significant impact on *Striga* emergence. A possible reason behind this low activity of EC formulated SL analogs is that the host crop was planted just after one week of the last application. Since EC formulated SL analogs are more stable so they may remain active in the soil for a longer period of time, which might continuously induce *Striga* germination that is easily attached to the host root. It is recommended that the host crop should not be grown just after the last application of EC-formulated SL analogs. We should consider enough time (3–4 months) or the whole rainy season for maximum induction of suicidal germination of *Striga* seeds. In spite of this, reduction in *Striga* infection led to better yield of both sorghum and pearl millet crops (Figure 8). In ICRISAT, Niger although it is an artificial infested field, we observed a huge variation among replicated plots (Figure 9). The emergence of *Striga* is limited such that was is hard to make conclusions from these findings. Low viability of *Striga* seeds, seed dormancy, cross-contamination among the plots, insect attack on the host, and poor growth of host plant can be possible reasons. We will repeat these field trials and expand our work in Kenya and Sudan to validate the findings, which will also expand our technology tackles different *Striga* ecotypes among Africa.

In summary, our results clearly demonstrate the effectiveness of the newly developed EC formulation of two potent and simple synthetic SL analogs as suicidal agents for practical field application. The new EC formulation of the two SL analogs appeared to be very bioactive, showing significant *Striga* reduction of 89–99% in rice under greenhouse conditions and 35–65% reduction in sorghum under field conditions in Burkina Faso. The reduction in *Striga* infection also led to increase rice plant height (61–71%) and pearl millet grain yield (>200%) as compared to blank treatment. The new EC formulation of the two SL analogs appeared to be very bioactive in terms of *Striga* reduction and host yield increase. In addition, the other advantages of EC formulation are simplicity, large-scale easy synthesis, stability, friendly use, easy packing/transportation and storage at normal room temperature. The development of proposed EC formulated simple SL analogs is the first step towards the large-scale synthesis of suicidal agents for field application and we believe this product will bring a breakthrough in suicidal technology to combat *Striga* in Africa.

## 4. Materials and Methods

### 4.1. Plant Materials and Chemicals

The SLs analogs MP3 and Nijmegen were synthesized and provided by Prof. Binne Zawanenburg, Radboud University, The Netherlands. Atlas G-1086 (provided by Croda Crop Care, Gouda, The Netherlands) was mixed with cyclo-hexanone (1:4) to make conventional formulation of SL analogs, used previously in the field. The new formulation “Emulsifiable Concentrate” (EC) of the two SL analogs (MP3, Nijmegen, The Netherlands) were prepared by UPL, India and Safety Data Sheet (SDS) has been attached (Appendix A). MP3 27EC means that 27 mg MP3 has been dissolved in 1.0 mL EC solvent (98.4 mM) while Nijmegen 34EC means that 34 mg Nijmegen has been added in 1.0 mL EC solvent (99 mM). *Striga hermonthica* seeds were collected from a sorghum (*Sorghum bicolor*) field during 2020 in Sudan (provided by Prof. A. G. Babiker), a maize (*Zea mays*) field during 2018 in Kenya (provided by Prof. Steven Runo, Kenyata University, Nairobi, Kenya), a pearl millet (*Pennisetum glaucum*) field during 2020 in Burkina Faso (provided by Dr. Djibril Yonli, INERA) and a pearl millet field during 2019 in Niger (provided by Dr. Mohammed Riyazaddin, ICRISAT, Niamey, Niger). Seeds of rice IAC-165 were obtained from Africa Rice, Tanzania (Courtesy of Dr. Jonne Rodenburg).

### 4.2. Striga Seed Germination Bioassays

First of all, the two formulations of both SL analogs were tested under lab conditions. *Striga* germination bioassays was conducted by following the procedure as described before [42]. The *Striga* seeds (collected from a sorghum, maize, and pearl millet infested field in Sudan, Kenya, Burkina Faso, and Niger, respectively) were first pre-conditioned before treating with SL analogs. For this purpose, the seeds were surface sterilized with 50% commercial bleach for 6–7 min and washed subsequently six times with MiliQ water in a laminar flow cabinet. The surface sterilized seeds (~50–100) were spread uniformly on a glass fiber filter paper disc (9 mm). Then 3 mL of sterilized MiliQ water was added in a plastic Petri plate containing one sterilized Whatman filter paper and 12 discs with *Striga* seeds. The Petri plates were sealed with parafilm, wrapped in aluminum foil, and incubated at 30 °C for 10 days. On the 11th day, the discs were dried in a laminar flow cabinet and the SL analogs (50 μL) were applied on each disc with various concentrations ranging from 10^−5^ M to 10^−11^ M. After application, the *Striga* seeds were induced to germinate in the dark for 24 h at 30 °C. The discs were scanned under a binocular microscope and germinated, and non-germinated seeds were counted by SeedQuant [43] and germination rate (in %) was calculated. 

### 4.3. Stability Analysis

The stability of both the formulations of two selected SL analogs was observed in an in vitro bioassay. Five filter papers were added in plastic Petri plates (9 cm). Then, 10 mL of MP3 or Nijmegen with EC- and AG formulations at 1.0 μM concentration were added in each Petri plate and sealed with parafilm to incubate in the dark at 30 °C for two-week intervals for up to 12 weeks. On the final week, five glass fiber filter paper discs, containing 50–100 pre-conditioned *Striga* seeds were added in each plate and incubated at 30 °C for 24 h. The discs were scanned under a binocular microscope and germinated and non-germinated seeds were counted by SeedQuant [43] and germination rate (in %) was calculated. 

### 4.4. Striga Emergence in Pots under Greenhouse Conditions

The biological activity of the two formulations of the SL analogs was further tested in pots under greenhouse conditions as described [25,32,44]. A sand and soil (Stender, Basissubstrat) mixture (1:3 ratio) was prepared. About 0.5 L of this mixture without *Striga* seeds was added in the bottom of a 3 L perforated plastic pot. Then about 20 mg (Approximately 8000) *Striga* seeds (collected from a sorghum field during 2020 in Sudan) were thoroughly mixed in 1.5 L soil mixture and added on the top of clean soil in the same pot. The pots were given light irrigation under greenhouse conditions at 30 °C for pre-conditioning of *Striga* seeds for 10 days. Then each pot was treated with 500 mL (1.0 μM) of various treatments to allow *Striga* seeds to germinate without host for another 10 days. Then one week old three rice seedlings (IAC-165) were planted in the middle of each pot. The rice plants were allowed to grow under normal growth condition (30 °C, 65% RH). After 70 days of sowing, *Striga* emergence was observed in each pot and compared with mock treatment. 

### 4.5. Striga Emergence under Mini-Box Conditions

The selected two SL analogs with both formulations were also evaluated in mini-boxes at INERA, Burkina Faso and at ICRISAT, Niger. At INERA, a 1 m × 1 m and 40 cm deep wooden box was used while in ICRISAT, Niger a mini-field, made of cemented bricks measuring 1 m × 2 m was used to assess *Striga* germination and emergence in response to SL treatments. The top 10 cm of soil in the box was infested with 50 mg of *Striga* seeds. Moreover, nine eplee bags containing surface sterilized *Striga* seeds (30–50 on average) were buried at a 10-cm depth in the box in an equidistant position and allowed them to pre-condition under hot and humid conditions for 10 days. At the end of the pre-conditioning of *Striga* seeds, both formulations of the SL analogs were applied (at 1.0 µM) and blank treatments (AG-1086 and EC) were included for comparison. Three eplee bags were taken out at 3, 6, and 9 days after SL application to count germinated and non-germinated seeds. After two weeks of application, pearl millet was sown in each box and emerged *Striga* plants were counted at 80 days after sowing (DAS).

### 4.6. Striga Emergence under Field Conditions

The two formulations of the candidate SL analogs were further assessed under naturally infested farmers field conditions in Eastern Burkina Faso and artificially infested field in ICRISAT, Niger. In Burkina Faso, two highly *Striga*-infested fields located near Kouaré research station (11°58′49″ N, 0°18′30″ E) of INERA were selected. In each field trial, the plots (4 × 4 m^2^) with various treatments were laid out by following randomized complete block design (RCBD) with six replications. Each plot comprised of five ridges spaced 80-cm apart. All of the plots were spaced with four (4) blank ridges to avoid any contamination of the treatments. The two formulations (EC vs. AG) of MP3 and Nijmegen were applied (25 mL/m^2^ at 400 µM) for six times, after onset of rainfall (≥10 mm) to make the final concentration of 1.0 µM. We included blank treatment as a control to compare the treatment effects. In the blank, we added the same amount of EC or AG emulsifier without SL analogs (active ingredients). Pearl millet (local cultivar Idipiéni) and sorghum (local cultivar Itchoari) crops were sown at 2 weeks after the last application. *Striga* emergence was counted at 110 days after crop planting. In ICRISAT, Niger the field was prepared with ploughing and planking and ridges with 80-cm spaces were made. Each plot consisted of 4 ridges (4 × 4 m^2^) and each ridge was infested with ~1.0 g *Striga* seeds. The trial was laid out by following RCBD with six replications. The field was artificially irrigated for pre-conditioning of *Striga* seeds for 10 days. All of the plots were treated with various treatments for six times and pearl millet (SOSTA-C88-P10) was sown at two weeks after the last application. *Striga* emergence was counted at 103 days after host planting.

### 4.7. Statistical Analysis

All of the data were collected by following standard procedure and collected data were analyzed statistically using statistical software package R (version 3.2.2). One-way analysis of variance (ANOVA) with Least Significant Difference (LSD) multiple range test and unpaired *t*-test were used for analyzing the effect of two formulations of the SL analogs on *Striga* infestation. 

## Figures and Tables

**Figure 1 plants-11-00808-f001:**
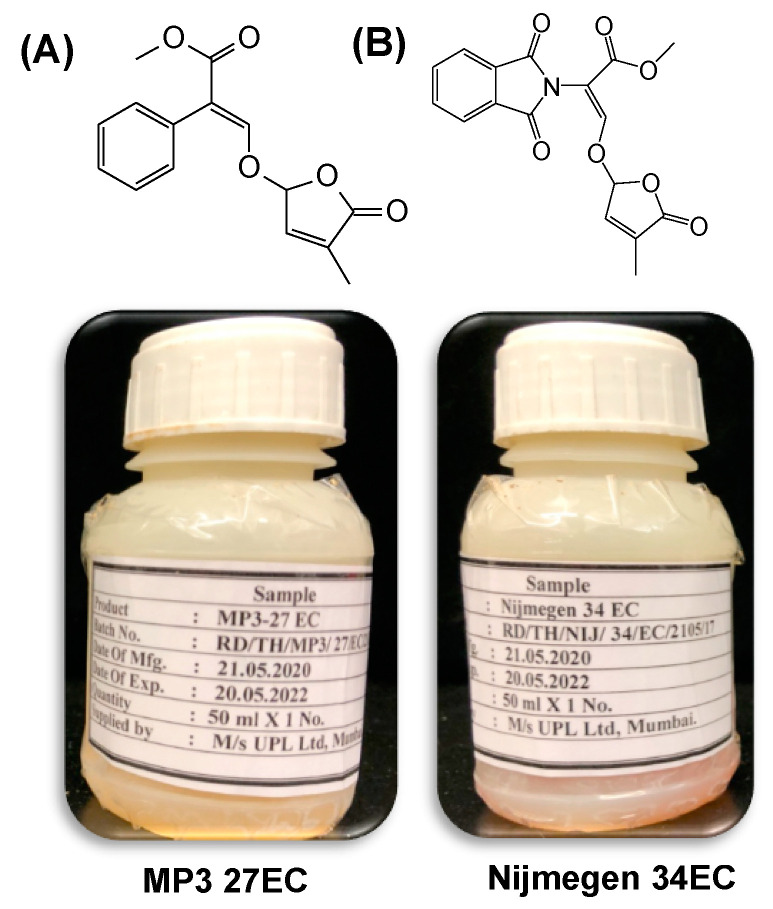
Structure of selected SL analogs and packaging of Emulsifiable Concentrate (EC) formulation of strigolactone analogs, developed by UPL, India (**A**) MP3 27EC and (**B**) Nijmegen 34EC, prepared by UPL, India.

**Figure 2 plants-11-00808-f002:**
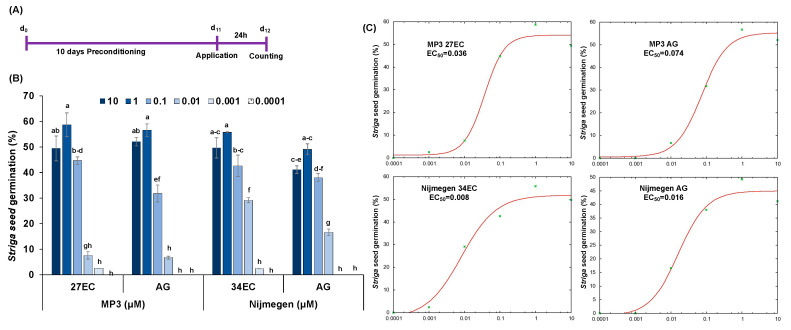
Comparison of EC formulated SL analogs MP3 27EC and Nijmegen 34EC with AG formulated SL analogs for *Striga* seed germination. (**A**) The scheme of the experiment conducted for *Striga* seed germination bioassays. (**B**) Effect of various concentrations of both formulations (AG and EC) of the two SL analogs on *Striga* seed germination. The *Striga* seeds collected from a sorghum infested field during 2020 in Sudan were treated with the two formulations of SL analogs. (**C**) EC_50_ of *Striga* seed germination in response to the various concentrations of the two SL analogs in different formulations. Data are means ± SE (*n* = 3), treatments with various letters differ significantly according to one-way analysis of variance (ANOVA) and Tukey’s post hoc test (*p* < 0.05).

**Figure 3 plants-11-00808-f003:**
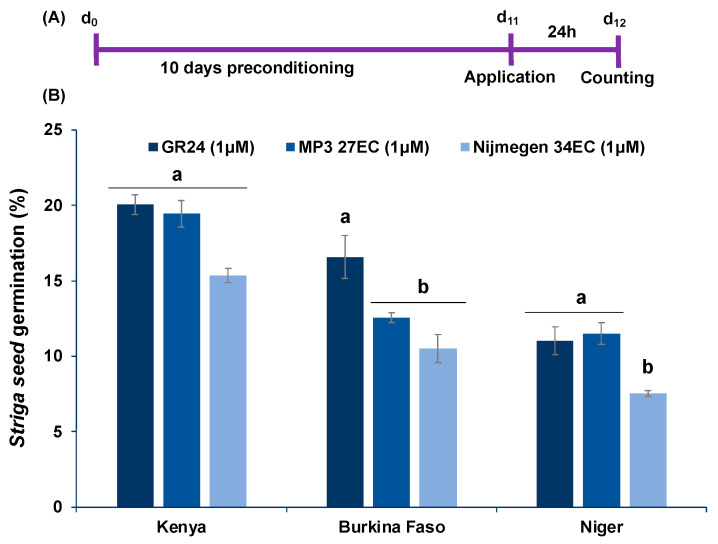
Effect of EC formulated SL analogs (MP3 27EC and Nijmegen 34EC) on *Striga* seed germination collected from a maize field in Kenya, pearl millet fields in Burkina Faso, and Niger. (**A**) Scheme of the experiment conducted for *Striga* seed germination bioassays. (**B**) *Striga* seed germination of various ecotype in response to MP3 27EC and Nijmegen 34EC. Data are means ± SE (*n* = 7). For each SL analog, treatments with various letters differ significantly according to one-way analysis of variance (ANOVA) and Tukey’s post hoc test (*p* < 0.05).

**Figure 4 plants-11-00808-f004:**
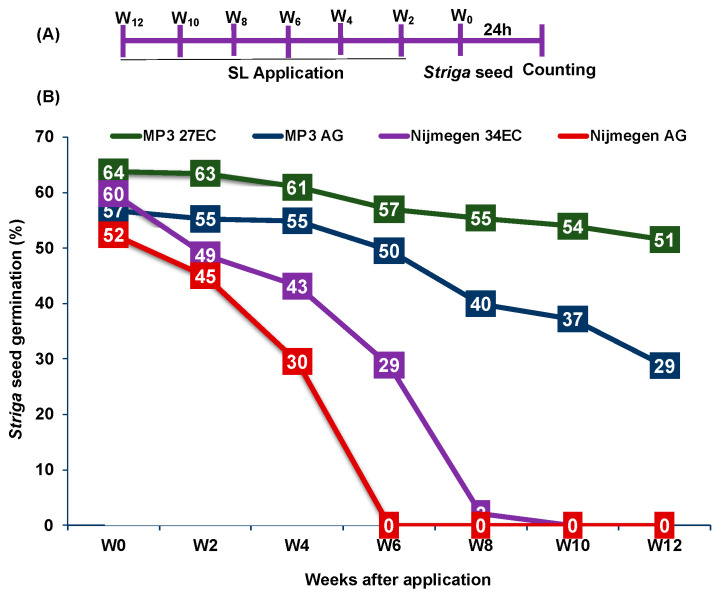
Stability of EC and AG formulation of the two strigolactone analogs. (**A**) Scheme of the experiment conducted biweekly to test the stability of SL analogs. (**B**) *Striga* seeds germination in response to two formulations of MP3 and Nijmegen at two-week intervals. Each SL analog (10 mL) was applied in a Petri plate on 5 filter papers and incubated at 30 °C for two-week intervals up to 12 weeks. The *Striga* seeds collected from a sorghum field in Sudan were evaluated for germination in each Petri plate. Data are means ± SE (*n* = 5).

**Figure 5 plants-11-00808-f005:**
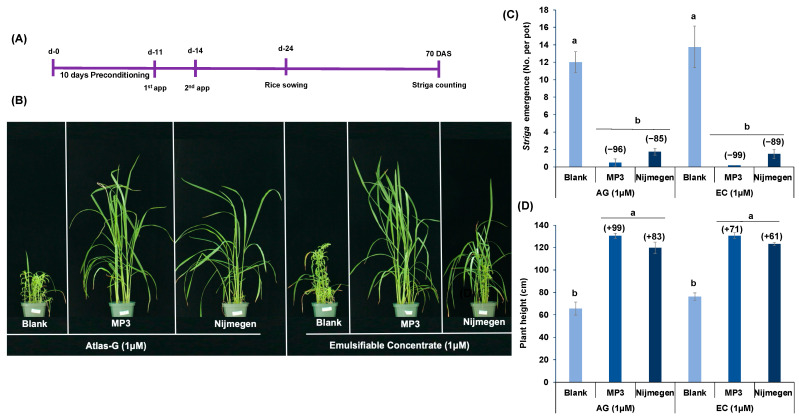
Effect of various formulations of strigolactone analogs on *Striga* emergence in pots under greenhouse conditions. (**A**) Scheme of the experiment conducted for *Striga* emergence in pots under greenhouse conditions. (**B**) View of the *Striga* infected pots. Each pot was filled with the soil infested with *Striga* seeds, collected from a sorghum field in Sudan. Both formulations of the two SL analogs (at 1.0 µM) were applied for two times in *Striga* infested pots. *Striga* emergence was counted at 70 days after sowing of rice. (**C**) Values of each bar showed average emergence of *Striga* per plot. (**D**) Average plant height of rice host plant measured at 70 DAS. Data are means ± SE (*n* = 4). For each SL analog, treatments with various letters differ significantly (*p* < 0.05). Values in parenthesis are showing the percentage increase (+) or decrease (−) over blank treatment.

**Figure 6 plants-11-00808-f006:**
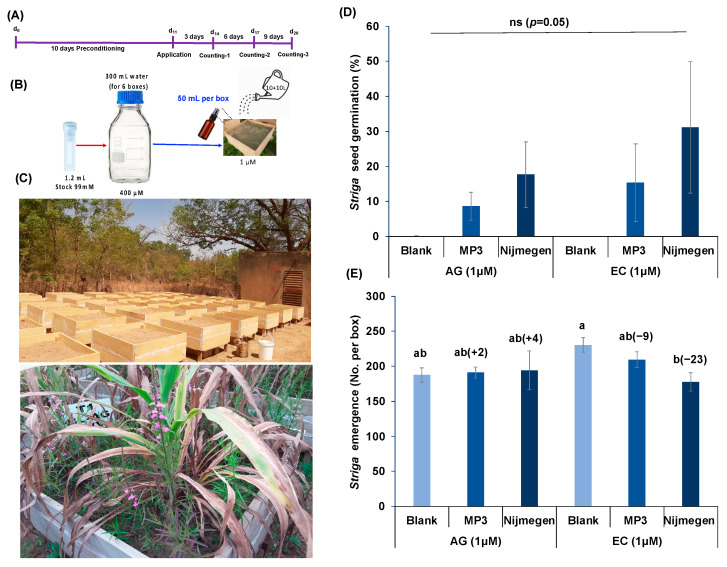
Effect of various formulations of SL analogs on *Striga* infection under mini-field conditions at Burkina Faso. (**A**) Scheme of the experiment conducted for *Striga* germination and emergence. (**B**) Experimental protocol adopted to conduct mini-field trials. Each box was filled with the soil, infested with *Striga* seeds collected from a pearl millet field in Burkina Faso. (**C**) View of the miniboxes and *Striga* infestation. (**D**) Average number of *Striga* seed germination in response to formulated MP3 and Nijmegen application. Both SL analogs of EC and AG formulations (at 1.0 µM) were applied in *Striga* infested mini-boxes. *Striga* germination was counted at 6 and 9 days after application. (**E**) Average number of *Striga* emergence in response to formulated MP3 and Nijmegen application. *Striga* emergence was counted after 80 days of pearl millet planting. Values of each bar showed average of *Striga* germination/emergence per minibox. Data are means ± SE (*n* = 6). For each SL analog, treatments with various letters differ significantly (*p* < 0.05). Values in parenthesis are showing the percentage increase (+) or decrease (−) over blank treatment. ns: non-significant.

**Figure 7 plants-11-00808-f007:**
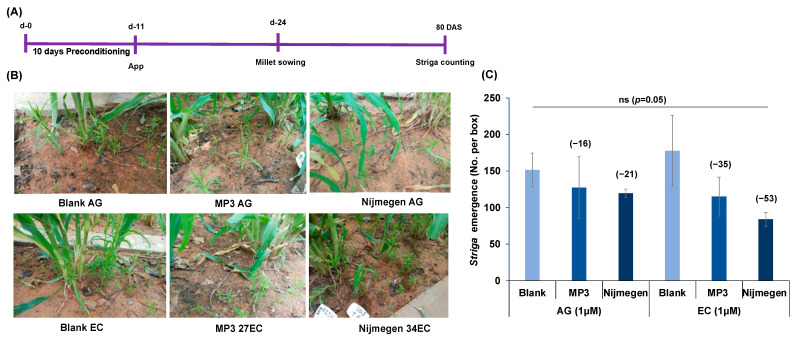
Effect of various formulations of SL analogs on *Striga* emergence under mini-field conditions at ICRISAT, Niger. (**A**) Scheme of the experiment conducted for *Striga* germination and emergence. (**B**) View of miniboxes and *Striga* infestation. Each box was filled with the soil, infested with *Striga* seeds collected from a pearl millet field in Niger. (**C**) Average number of *Striga* emergence in response to formulated MP3 and Nijmegen application. *Striga* emergence was counted after 80 days of pearl millet planting. Values of each bar showed average of *Striga* emergence per minibox. Data are means ± SE (*n* = 3). Values in parenthesis are showing the percentage increase (+) or decrease (−) over blank treatment. ns: non-significant.

**Figure 8 plants-11-00808-f008:**
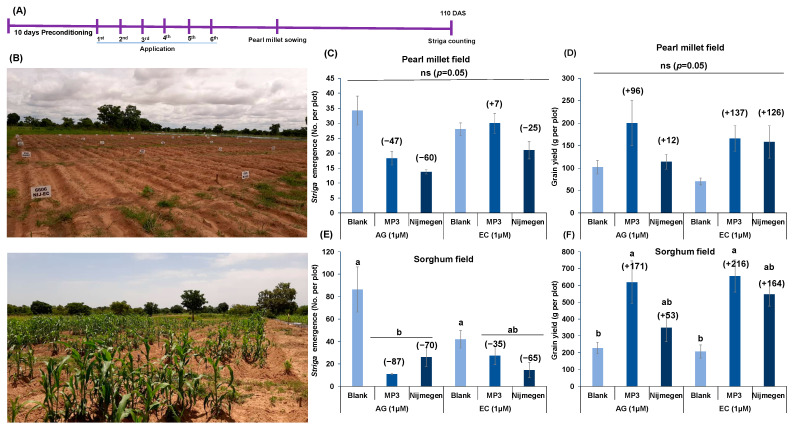
Effect of various formulations of SL analogs on *Striga* emergence and grain yield under naturally infested farmers field conditions at INERA, Burkina Faso. (**A**) Scheme of the experiment conducted for *Striga* emergence under farmers field conditions during 2020. (**B**) View of the *Striga* infested farmers field at Fada N’gourma, Burkina Faso. (**C**) Average number of *Striga* emergence in the pearl millet field in response to formulated MP3 and Nijmegen application. (**D**) Average grain yield per plot from the pearl millet field in response to formulated MP3 and Nijmegen application. (**E**) Average number of *Striga* emergence in the sorghum field in response to formulated MP3 and Nijmegen application. (**F**) Average grain yield per plot from the sorghum field in response to formulated MP3 and Nijmegen application. Both SL analogs of EC and AG formulations (at 1.0 µM) were applied in the *Striga* infested pearl millet and sorghum farmer fields. *Striga* emergence was counted after 110 days of pearl millet and sorghum planting. Values of each bar showed average of *Striga* emergence per plot. Data are means ± SE (*n* = 4). For each SL analog, treatments with various letters differ significantly (*p* < 0.05). Values in parenthesis are showing the percentage increase (+) or decrease (−) over blank treatment. ns: non-significant.

**Figure 9 plants-11-00808-f009:**
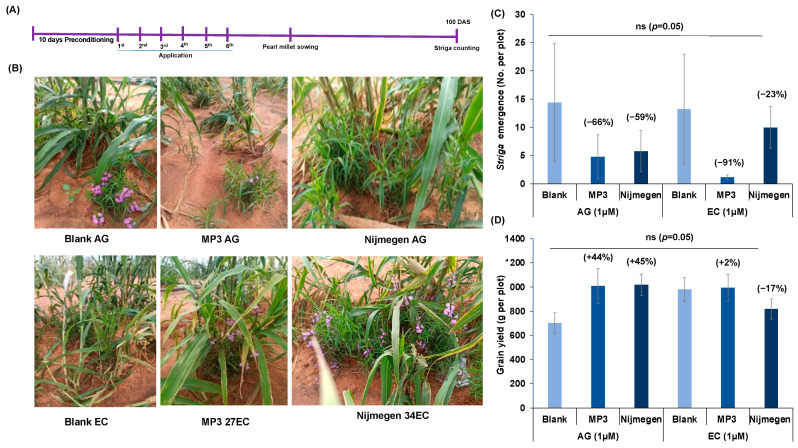
Effect of EC and AG formulations of the two SL analogs (MP3 and Nijmegen) on *Striga* emergence under artificially infested field conditions at ICRISAT, Niger. (**A**) Scheme of the experiment conducted for *Striga* emergence under field conditions. (**B**) View of the *Striga* infested farms under various treatments during 2020. (**C**) Average number of *Striga* emergence in the pearl millet field in response to formulated MP3 and Nijmegen application. (**D**) Average grain yield per plot from the pearl millet field in response to formulated MP3 and Nijmegen application. Both SL analogs of EC and AG formulations (at 1.0 µM) were applied in the *Striga* infested pearl millet field. *Striga* emergence was counted after 103 days of pearl millet planting. Values of each bar showed average of *Striga* emergence per plot. Data are means ± SE (*n* = 5). Values in parenthesis are showing the percentage increase (+) or decrease (−) over blank treatment. ns: non-significant.

## Data Availability

Not applicable.

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
