# Peer review of "A New Formulation for Strigolactone Suicidal Germination Agents, towards Successful Striga Management"

_plants, 2022, doi:10.3390/plants11060808_

Round 1
Reviewer 1 Report
the manuscript "A New Formulation for Strigolactone Suicidal Germination Agents, Towards Successful Striga Management" cover a very important problem with Stringa (not only in Africa BUT and for Europe).
The only part where authors must to improve is the conclusion (make focus in results).
The second point is the references format
Author Response
Dear Reviewer
Thank you for reviewing our manuscript entitled “A New Formulation for Strigolactone Suicidal Germination Agents, Towards Successful Striga Management” by Jamil et al. for publication in the PLANTS.
We have addressed and incorporated all the suggestions raised.
Q1: The only part where authors must to improve is the conclusion (make focus in results).
Response: We thank the reviewer for the suggestion. We have improved the conclusion as suggested.
Q2: The second point is the references format.
Response: We have corrected all references as per Journal format.
Reviewer 2 Report
Striga hermonthica is a severe root-parasitic weed infesting several cereal crops. legume crops and sugarcane in many countries of Africa and Asia, as well as in Australia. Specially, in African countries, striga infestation in agriculture causing threat to the life and food security of 300 million people. The management of striga is difficult due to the huge seed production (0.1 million seeds per plant), longer seed dormancy (up to 20 years), complex life cycle, underground damage, and host dependency of striga seed germination. Comprehensive and sustainable control measures of striga are not yet available in the world. As we know that germination of striga seed requires host-released chemical signals (such as strigolactones (SLs)) and the germinated seeds should establish a connection to the root system of the host to survive due to very limited food reserve in tiny seeds for a short period of time. Therefore, the main strategy to control this weed is to induce seed germination by using stable stimulants, and do not allow germinated seeds to make connection to the host plants.
The SLs analogs MP3 and Nijmegen were synthesized by Radboud University, The Netherlands, and two new emulsifiable concentrate (EC) formulation of MP3 and Nijmegen were prepared by UPL, Indian. The authors conducted a series of experiments to evaluate efficacy of the newly developed EC formulation of MP3 and Nijmegen on seed germination of African striga ecotypes under laboratory, greenhouse, and field conditions in different African countries. All the experimental procedures of this study were taken considering weed biology and ecology, soil types, common agronomic practices for crop production, climatic conditions, and chemical interaction with the plant. Data presented in this study were reliable, sufficient, and potential to develop such the EC formulation of MP3 and Nijmegen. The authors clearly demonstrated the effectiveness of the newly developed EC formulation of two potent and simple synthetic SL analogs as suicidal agents for practical field application. The new EC-formulation of the two SL analogs appeared to be very bioactive in terms of striga reduction and host yield increase. In addition, advantages of EC-formulation are simplicity, large-scale easy synthesis, stability, friendly use, easy packing/transportation and storage at normal room temperature. I believe that the authors successfully developed a desired formulation for applying SL analogs as suicidal agents for sustainable striga management strategies in African agriculture. This manuscript is potential worldwide for controlling striga weed in crop production and extending knowledge in weed science and product development. I am gladly recommending this manuscript for publication in the plants.
Comments
Caption for the Figure 6 is not completed according to the figures A, B, C, D and E; please make necessary corrections.
Author Response
Dear Reviewer
Thank you for reviewing our manuscript entitled “A New Formulation for Strigolactone Suicidal Germination Agents, Towards Successful Striga Management” by Jamil et al. for publication in the PLANTS.
We have addressed and incorporated all the suggestions raised.
Q1: Caption for the Figure 6 is not completed according to the figures A, B, C, D and E; please make necessary corrections.
Response: We have made necessary corrections in Figure 6.
Reviewer 3 Report
This manuscript reported a new formulation for two previously reported strigolactone analogs towards Striga management. As the authors described in this manuscript, proper formulation for field application is critical to the success of the suicidal germination by strigolactone analogs. The authors examined the effects of strigolactone analogs, MP3 and Nijmegen in laboratory or field levels, using emulsifiable concentrate (EC) formulation. The EC formulation increased the stability of strigolactone analogs, reduced Striga infestation to sorghum and pearl millet, and recovered crop yield in the fields of Burkina Faso and Niger. These results could contribute to the development of suicidal germination technology. The reviewer would like to recommend the acceptance of this manuscript after addressing the following points.
- Figure 4. The authors tested the stability of EC formulated SLs at the laboratory level, found that EC formulations are more stable than Atlas-G formulations. What attributes this difference? The stability of strigolactones depend on pH condition. What is the pH of each formulation?
- Figure 6. Figure legend is inadequate. The reviewer could not find a legend for Fig. 6E.
- Figures 6-9. What was treated in the blanks of EC and Atlas-G formulation? Please describe in detail. If nothing was treated, why are the results of each Blank so different?
- Materials and methods section. Please describe the varieties of sorghum and pearl millet used in this manuscript.
- Does AG mean Atlas-G? If so, the notation in the text and figures do not match. Please correct.
Author Response
Dear Reviewer
Thank you for reviewing our manuscript entitled “A New Formulation for Strigolactone Suicidal Germination Agents, Towards Successful Striga Management” by Jamil et al. for publication in the PLANTS.
We have addressed and incorporated all the suggestions raised.
Q1: Figure 4. The authors tested the stability of EC formulated SLs at the laboratory level, found that EC formulations are more stable than Atlas-G formulations. What attributes this difference? The stability of strigolactones depend on pH condition. What is the pH of each formulation?
Response: We are thankful to the reviewer for this valuable comment. We tested the stability and activity of EC-formulated SL analogs under varying values of pH. We observed more stability and better activity of EC formulated SLs under low pH (acidic) as compared to high pH (data not shown). The pH of African soil is also low (acidic) that is very suitable for EC-formulated SL analogs.
Q2: Figure 6. Figure legend is inadequate. The reviewer could not find a legend for Fig. 6E.
Response: Thanks to reviewer. We have corrected the legend for Fig. 6E.
Q3: Figures 6-9. What was treated in the blanks of EC and Atlas-G formulation? Please describe in detail. If nothing was treated, why are the results of each Blank so different?
Response: We have added this detail in the methods Part: “We included blank treatment as a control to compare the treatment effects. In blank we added same amount of EC or AG without SL analogs (active ingredients).”
Materials and methods section.
Q4: Please describe the varieties of sorghum and pearl millet used in this manuscript.
Response: We have described varieties of sorghum and pearl millet used in this manuscript.
Q5: Does AG mean Atlas-G? If so, the notation in the text and figures do not match. Please correct.
Response: We have corrected as suggested.